# Genomic insights into the expansion of meropenem-resistant GPSC1-CC320 *Streptococcus pneumoniae* serotype 19A isolates from children under 5 years of age with invasive infections, 2018–2024

Jin Lee[1], Jungsun Park[1], Tae-Min La[2], Ji-Yeon Hyeon[2], Hwanhee Kim[1], Seo Yeon Ko[1], Dong Hyeok Kim[1], Jaeil Yoo[1], Junyoung Kim [1]*

1 Division of Bacterial Diseases, Department of Laboratory Diagnosis and Analysis, Korea Disease Control and Prevention Agency, Cheongju, South Korea, 2 College of Veterinary Medicine, Konkuk University, Seoul, South Korea

* jun49@hanmail.net

## Abstract

The global rise of multidrug-resistant (MDR) *Streptococcus pneumoniae* serotype 19A despite widespread pneumococcal conjugate vaccine (PCV) use poses an emerging challenge for pediatric infectious disease control. In South Korea, the detection of serotype 19A initially declined following PCV13 implementation, but it has recently re-emerged, raising concerns over the evolution and persistence of high-risk clones.

We investigated invasive serotype 19A isolates recovered from children under five years of age between 2018 and 2024. Antimicrobial susceptibility testing and whole-genome sequencing (WGS) were conducted to characterize resistance mechanisms, clonal structure, and phylogenetic relationships.

Among the 104 invasive pneumococcal isolates, 11 (10.6%) were identified as serotype 19A. All of these belonged to GPSC1/CC320 and were classified as ST320 or ST6400. These isolates exhibited resistance to meropenem but retained susceptibility to cefotaxime, carrying a conserved pbp1a-13, pbp2b-11, and pbp2x-16 combination. Phylogenetic analysis revealed four distinct subclades among the South Korean isolates, each showing high similarity to international strains.

To our knowledge, this is the first genomic study of meropenem-resistant 19A in children from South Korea. Our findings suggest that the resurgence of 19A is driven by stable, high-risk MDR. These results underscore the need for clone-level genomic surveillance to preempt emerging threats and inform next-generation vaccine and antibiotic strategies.

## Introduction

*Streptococcus pneumoniae* is a major cause of invasive pneumococcal disease (IPD), including bacteremia, meningitis, and osteomyelitis, and is responsible for

**Data availability statement:** All whole-genome sequencing data have been deposited in the NCBI Sequence Read Archive under BioProject accession number PRJNA1250463 (SRA accessions: SRX28386539–SRX28386549).

**Funding:** This work was supported by a grant from the Korea Disease Control and Prevention Agency (grant number 6331-301-210). The funders had no role in study design, data collection and analysis, decision to publish, or preparation of the manuscript.

**Competing interests:** The authors have declared that no competing interests exist.

substantial morbidity and mortality among children worldwide [1,2]. According to the World Health Organization (WHO), IPD leads to nearly one million deaths annually in children under five years of age, underscoring its global public health significance [2]. In India alone, a national surveillance study reported a high burden of pneumococcal disease in children under five years of age [1].

To reduce the burden of IPD, many countries introduced pneumococcal conjugate vaccines (PCVs), beginning with the 7-valent formulation (PCV7), which led to a marked reduction in IPD caused by vaccine-included serotypes. However, serotype replacement, most notably the global rise of serotype 19A, has emerged as a critical issue [3,4]. In response, PCV10 and PCV13 were developed, with PCV13 uniquely including serotype 19A. Although PCV13 implementation significantly reduces the disease burden associated with this serotype, residual IPD caused by 19A continues to be reported, even in countries with high vaccine uptake [5].

A global surveillance analysis by the Pneumococcal Serotype Replacement and Distribution Estimation (PSERENADE) project recently confirmed that serotype 19A remains one of the predominant causes of post-PCV IPD worldwide [6]. Additionally, breakthrough infections with 19A have been documented among children who were appropriately vaccinated with PCV13, suggesting that vaccine-driven selective pressure may not fully eliminate high-risk clones [7].

The persistence of serotype 19A has been closely linked to the global expansion of sequence type (ST) 320, a multidrug-resistant lineage within clonal complex 320 (CC320) and global pneumococcal sequence cluster 1 (GPSC1) [8,9]. This clone is resistant to penicillin, macrolides, cephalosporins, and, increasingly, carbapenems and has been widely detected across the United States, Asia, and Europe [8–10]. Phylogeographic studies in Japan and other regions have shown sublineage diversification of ST320 clones with region-specific adaptations, including variations in β-lactam resistance-associated pbp alleles and recombination events leading to capsular switching [8,10].

In South Korea, PCV7 was introduced in 2003 as an optional vaccine for children under five years of age and was replaced by PCV10 or PCV13 in 2010. These vaccines were incorporated into the National Immunization Program (NIP) in 2014. According to the latest national report, PCV coverage among children in South Korea reached 97% in 2023 [11]. In April 2024, PCV10 was discontinued, and PCV15 was introduced alongside PCV13. Surveillance data revealed an initial increase in serotype 19A following PCV7, followed by a decrease after PCV13 implementation [12]. However, recent data indicate the possible resurgence of 19A, particularly in children under two years of age. Most isolates identified in South Korea belong to the ST320 lineage and exhibit multidrug resistance [13]. These observations raise important questions about the evolutionary trajectory, antimicrobial resistance mechanisms, and transmission dynamics of circulating serotype 19A clones. Understanding the genetic structure of these strains under current vaccine pressure is essential for predicting future changes in pneumococcal population dynamics.

This study aimed to characterize *S. pneumoniae* serotype 19A isolates recovered from children under five years of age in South Korea between 2018 and 2024. Using

whole-genome sequencing (WGS), we analyzed their clonal composition, resistance profiles, and phylogenetic relationships to assess the genomic factors contributing to their re-emergence.

## Methods

### Serotype distribution

*S. pneumoniae* isolates in South Korea were collected through the Infectious Diseases Surveillance Systems operated by the Korea Disease Control and Prevention Agency (KDCA). In this study, a total of 104 isolates collected from children under 5 years of age between 2018 and 2024 were confirmed as *S. pneumoniae*, and their serotypes were determined through serotyping analysis. The annual distribution of the isolates was 2018 (22), 2019 (10), 2020 (7), 2021 (13), 2022 (15), 2023 (18), and 2024 (19). Among them, 11 isolates identified as serotype 19A were analyzed separately.

### *Streptococcus pneumoniae* identification and serotyping

The isolated *S. pneumoniae* strains were cultured on 5% sheep blood agar plates incubated at 37°C under 5% $CO_2$ for 18–20 h. *S. pneumoniae* identification was performed according to colony morphology, alpha-hemolysis activity, Gram staining, catalase activity, and optochin susceptibility. After confirming the identification of *S. pneumoniae*, the isolates were stored in Microbank vials (Pro-lab Diagnostics, Richmond Hill, Canada) at −80°C until further testing. All the isolates were serotyped by the Quellung reaction using antisera (Statens Serum Institut, Copenhagen, Denmark). After the Quellung reaction, additional serotype detection was confirmed by conventional multiplex PCR protocols with serogroup/serotype primers provided by the CDC (http://www.cdc.gov./streplab.pcr.htm). A primer pair targeting the *cps*A gene was used as a positive control for each reaction. The PCV13 serotypes were serotypes 1, 3, 4, 5, 6A, 7F, 9V, 14, 18C, 19A, 19F, and 23F, and the non-PCV13 serotypes included all other serotypes.

### Antibiotic susceptibility testing of the 19A strain

Antimicrobial susceptibility was determined via the broth microdilution method using a Sensititre IB-VPD Panel (Thermo Scientific, Waltham, MA, USA) by measuring the minimum inhibitory concentrations for serotype 19A isolates within different concentration ranges. The following 16 antimicrobial agents were assayed: penicillin, amoxicillin/clavulanic acid (at a 2:1 ratio), meropenem, cefotaxime, cefuroxime, ceftriaxone, cefepime, azithromycin, erythromycin, tetracycline, levofloxacin, trimethoprim/sulfamethoxazole, clindamycin, chloramphenicol, rifampin, and vancomycin. The results were interpreted according to the Clinical and Laboratory Standards Institute (CLSI) criteria (M100, 28th edition). *S. pneumoniae* ATCC49619 was used as a quality control strain. The results were interpreted according to the CLSI-recommended breakpoints. Among the 11 serotype 19A isolates, one isolate was recovered from cerebrospinal fluid (CSF), and the remaining ten were from non-CSF sources. For the single isolate obtained from CSF, meningitis breakpoints were applied, whereas for isolates from all other sources, nonmeningitis breakpoints were used.

### Whole-genome sequencing

Genomic DNA was extracted from each sample using the DNeasy Blood and Tissue Kit (Qiagen, Germany). The quality of the purified total DNA was measured using a NanoDrop 2000 spectrophotometer (Thermo Fisher, Waltham, MA, USA). The concentration was determined with a Qubit 4 fluorometer using a high-sensitivity kit (Invitrogen, Waltham, MA, USA). Library fragment lengths were assessed through the use of a Bioanalyzer TapeStation with a DNA 1000 Kit (Agilent Technologies, Inc., Santa Clara, CA, USA). Genomic libraries were prepared via an Illumina DNA Prep Kit according to the manufacturer's protocol. Sequencing was performed using a 300-cycle (2 × 150 bp paired-end) MiSeq reagent kit, version 3, with a MiSeq sequencer. The raw sequences generated by Illumina MiSeq were quality filtered using FastQC, with the average quality set at Q30. Raw reads from Illumina sequencing were quality trimmed and assembled using CLC Genomics Workbench 24 (Qiagen, Hilden, Germany).

### Genetic characterization

The assembled sequences were analyzed for antimicrobial resistance genes (ResFinder 4.6) using web tools available from the Center for Genomic Epidemiology (CGE) (http://www.genomicepidemiology.org/). Genes encoding virulence factors were identified using the virulence factor database (VFDB, http://www.mgc.ac.cn/VFs/). All the isolates were analyzed for their penicillin-binding protein (PBP) signature, clonal complexes (CCs) from the STs and global pneumococcal sequence clusters (GPSCs) by uploading assembled genomes to Pathogenwatch (http://pathogen.watch/). The population structure was defined by assigning a GPSC to each isolate using a kmer-based clustering method, PopPUNK [14], and a reference list of pneumococcal isolates in the GPS database (https://www.pneumogen.net/gps/assigningGPSCs.html). The analysis results can be viewed online via MicroReact (http://microreact.org/).

### Ethics statement

This study used archived *S. pneumoniae* isolates and associated metadata collected through the KDCA's routine national surveillance system between 2018 and 2024. All isolates were obtained for public health surveillance purposes, and all associated data were fully anonymized before being accessed by researchers. As the study involved analysis of anonymized, non-identifiable archived isolates collected under a public mandate, Institutional Review Board (IRB) approval and individual informed consent were not required, in accordance with the guidelines of the KDCA Institutional Review Board.

### Nucleotide sequence accession numbers

The whole-genome sequences of these strains were deposited with the National Center for Biotechnology Information (NCBI) under BioProject accession number PRJNA1250463 (SRA accessions: SRX28386539–SRX28386549).

## Results

### Serotype distribution

A total of 104 *S. pneumoniae* isolates causing IPD were recovered from children under 5 years of age and subsequently serotyped. Among these isolates, 15 (14.4%) were identified as PCV13 serotypes. Serotype 19A was the most prevalent (10.6%), followed by serotype 19F (3.8%). A total of 89 isolates (85.6%) were classified as non-PCV13 serotypes, with serotypes 10A, 15C, 15B, and 23B being the most frequently identified (Table 1). The proportion of PCV13 serotypes has markedly increased in recent years, increasing from 9.1% (2/22) in 2018 to 27.8% (5/18) in 2023 and reaching 31.6% (6/19) in 2024. Notably, serotype 19A was the most predominant PCV13 serotype, with its detection rate increasing from 9.1% (2/22) in (2018) and 14.3% (1/7) in (2020) to 16.7% (3/18) in (2023) and 26.3% (5/19) in (2024). Furthermore, a comparison between the two periods (2018–2022 vs. 2023–2024) demonstrated significant increases in both the overall proportion of PCV13 serotypes (p = 0.0023) and serotype 19A specifically (p = 0.0154) (Fig 1).

### Antimicrobial susceptibility of 19A

The antimicrobial susceptibility profiles of eleven *S. pneumoniae* serotype 19A isolates were tested against 16 antibiotics according to the latest CLSI breakpoints are summarized in Table 2. All 11 isolates were resistant to tetracycline, azithromycin, cefuroxime, meropenem, trimethoprim/sulfamethoxazole, amoxicillin/clavulanic acid, and erythromycin but remained susceptible to vancomycin, rifampin, levofloxacin, and chloramphenicol. Several isolates exhibited intermediate resistance to β-lactam antibiotics, including penicillin, cefotaxime, cefepime, and ceftriaxone.

### Genetic characterization of 19A

The observed resistance of the 19A isolates was confirmed by identifying the presence of the *tet*(M) gene encoding tetracycline resistance and the *mef*(A), *erm*(B) and *msr*(D) genes encoding macrolide resistance. All the isolates presented

**Table 1. Serotype distribution among 104 *S. pneumoniae* isolates from children <5 years of age with IPD in South Korea, 2018–2024.**

| Serotype | Number (%) |
|---|---|
| **PCV13** | **15 (14.4)** |
| 19A | 11 (10.6) |
| 19F | 4 (3.8) |
| **non-PCV13** | **89 (85.6)** |
| 10A | 23 (22.1) |
| 15C | 14 (13.5) |
| 15B | 13 (12.5) |
| 23B | 13 (12.5) |
| 15A | 6 (5.8) |
| 35B | 5 (4.8) |
| 23A | 4 (3.8) |
| 34 | 4 (3.8) |
| 22F | 1 (1.0) |
| 11A | 1 (1.0) |
| 12F | 1 (1.0) |
| 38 | 1 (1.0) |
| 24F | 1 (1.0) |
| 6C | 1 (1.0) |
| Non-typeable | 1 (1.0) |
| Total | 104 (100.0) |

the PBP type combination 13 (PBP1a)-11(PBP2b)-16(PBP2x). The intermediate resistance or full resistance to penicillin, cefotaxime, ceftriaxone and cefepime can be explained by the presence of a mutation in the *pbp* genes [15]. The 19A isolates belonged to GPSC1 and the CC320 clonal complex with ST6400 and ST320 (Table 2). Among the virulence factors, choline-binding protein-encoding genes, including *cbpE*, *cbpG*, *lytA*, *lytB*, *lytC*, *pre/cbpE*, and *pspA,* were identified in all the isolates, whereas *pspC/cbpA* was absent in some. Genes encoding fibronectin-binding protein (*pavA*), laminin-binding protein (*lmb*), streptococcal lipoprotein rotamase A (*slrA*), streptococcal plasmin receptor/GAPDH (*plr/gapA*), and rlrA islet components (*rrgA*, *rrgB*, *rrgC*, *srtB*, *srtC*, and *srtD*) were also consistently detected. In addition, a set of other virulence-associated genes was identified, including *hysA* (hyaluronidase), *nanA* (neuraminidase A), *eno* (enolase), *piaA* and *piuA* (iron uptake), *psaA* (manganese uptake), protease-related genes (*cppA*, *iga*, *htrA/degP*, *tig/ropA*, *zmpB*), and *ply* (pneumolysin) (Table 3).

## GPSC1 representation in the GPS global dataset

A total of 1300 isolates belonging to GPSC1 were identified in the GPS global dataset from 1994 through 2024 (Fig 2). The GPSC1 isolates identified in this study were analyzed in an international context using GPSC1 from the GPS collection, which contained serotypes 3, 6C, 14, 19A, 19F and 23F. The majority of the GPSC1 isolates were from the USA (28%, 366/1300), followed by Thailand (27%, 359/1300), South Africa (10%, 134/1300), China (7%, 87/1300) and Japan (4%, 56/1300). Serotype 19A represented 40.8% (535/1300) of the GPSC1 isolates. The international GPSC1 phylogeny revealed the presence of four distinct subclades among the South Korean GPSC1 serotype 19A isolates identified in this study. Each subclade shared a common ancestor with isolates from other geographical regions diverging within the South Korean lineage. For example, all six isolates of ST6400 clustered closely together, whereas the remaining five isolates of ST320 were dispersed across the international GPSC1 phylogenetic tree. Strain 20–1 was closely related to the isolate

**Fig 1. Distribution of PCV13 and non-PCV13 serotypes among invasive *S. pneumoniae* isolates from South Korean children under 5 years of age, 2018–2024. (A)** Annual distribution of serotype 19A among 104 *S. pneumoniae* isolates from children <5 years of age with IPD in South Korea, 2018–2024. **(B)** Comparison of PCV13 versus non-PCV13 isolates (left) and serotype 19A versus non-19A isolates (right) between the periods 2018–2022 and 2023–2024. Statistical significance was determined using Fisher's exact test (left: p = 0.0023; right: p = 0.0154).

collected in Peru in 2011 and South Korea in 2012. Strain 23–71 belonged to the same subclade as isolates reported in Japan between 2009 and 2014. Additionally, strains 24–43 and 24–175 were closely related to isolates collected in Canada in 2014, whereas strain 24–243 was related to isolates collected in the USA in 2016.

## Discussion

In this study, we performed WGS of meropenem-resistant *S. pneumoniae* serotype 19A isolates collected from pediatric IPD patients under five years of age in South Korea between 2018 and 2024. These isolates, belonging to the GPSC1–CC320 clonal complex, were analyzed to elucidate their resistance mechanisms and genomic features associated with their re-emergence in a population with high PCV13 coverage.

Following the introduction of PCV10 and PCV13 into the Korean National Immunization Program in 2014 and the transition to exclusive PCV13 use in 2020, the incidence of serotype 19A IPD declined markedly. Notably, serotype 19A was not detected in children under five years of age between 2021 and 2022. However, 19A re-emerged in 2023 and showed a sharp increase in 2024. A substantial proportion of the isolates were recovered from infants under 24 months of age, suggesting age-specific susceptibility or clonal spread that merits further investigation [16–18].

All the isolates in this study belonged to GPSC1 and the CC320 clonal complex and consisted of two sequence types: ST320 and ST6400. ST320, first characterized in Asia, including South Korea, has become a globally disseminated

**Table 2. Comparison of phenotypic and WGS-derived antimicrobial resistance profiles of serotype 19A along with their MLST profiles.**

| Characteristic | | Isolates | | | | | | | | | | |
|---|---|---|---|---|---|---|---|---|---|---|---|---|
| | | 18-85 | 18-185 | 20-1 | 23-4 | 23-71 | 23-113 | 24-43 | 24-175 | 24-189 | 24-243 | 24-250 |
| Phenotypic | MIC interpretation | | | | | | | | | | | |
| | Penicillin | I | I | I | R | I | R | I | R | R | R | I |
| | Amoxicillin/ Clavulanic Acid 2:1 | R | R | R | R | R | R | R | R | R | R | R |
| | Ceftriaxone | I | I | I | I | I | I | I | R | I | R | I |
| | Cefotaxime | I | I | I | I | I | I | I | R | I | I | I |
| | Cefepime | I | I | I | I | I | I | I | R | I | R | I |
| | Cefuroxime | R | R | R | R | R | R | R | R | R | R | R |
| | Meropenem | R | R | R | R | I | R | R | R | R | R | R |
| | Erythromycin | R | R | R | R | R | R | R | R | R | R | R |
| | Azithromycin | R | R | R | R | R | R | R | R | R | R | R |
| | Tetracycline | R | R | R | R | R | R | R | R | R | R | R |
| | Trimethoprim/ Sulfamethoxazole | R | R | R | R | R | R | R | R | R | R | R |
| | Clindamycin | S | S | S | S | R | S | R | R | S | S | S |
| | Levofloxacin | S | S | S | S | S | S | S | S | S | S | S |
| | Chloramphenicol | S | S | S | S | S | S | S | S | S | S | S |
| | Rifampin | S | S | S | S | S | S | S | S | S | S | S |
| | Vancomycin | S | S | S | S | S | S | S | S | S | S | S |
| Genotypic antibiotic resistance | Penicillin resistance | | | | | | | | | | | |
| | PBP1a | 13 | 13 | 13 | 13 | 13 | 13 | 13 | 13 | 13 | 13 | 13 |
| | PBP2b | 11 | 11 | 11 | 11 | 11 | 11 | 11 | 11 | 11 | 11 | 11 |
| | PBP2x | 16 | 16 | 16 | 16 | 16 | 16 | 16 | 16 | 16 | 16 | 16 |
| | Other antibiotics | | | | | | | | | | | |
| | erm(B) | + | + | + | + | + | + | + | + | + | + | + |
| | mef(A) | + | + | + | + | + | + | + | + | + | + | + |
| | msr(D) | + | + | + | + | + | + | + | + | + | + | + |
| | tet(M) | + | + | + | + | + | + | + | + | + | + | + |
| MLST | ST(CC) | 6400 (320) | 6400 (320) | 320 (320) | 6400 (320) | 320 (320) | 6400 (320) | 320 (320) | 320 (320) | 6400 (320) | 320 (320) | 6400 (320) |
| GPSC | | 1 | 1 | 1 | 1 | 1 | 1 | 1 | 1 | 1 | 1 | 1 |

MIC, minimum inhibitory concentration; S, susceptible; I, intermediate; R, resistant; MLST, multilocus sequence typing; ST, sequence type; CC, clonal complex; GPSC, global pneumococcal sequence clusters.

multidrug-resistant (MDR) clone with high invasive potential [9,19,20]. ST6400, first reported domestically in 2010, was identified at a comparable frequency in recent isolates, indicating ongoing diversification within CC320. This observation aligns with sublineage expansions previously documented in Japan [8]. Furthermore, our phylogenetic analysis demonstrated that South Korean 19A isolates formed four distinct subclades interspersed within international ST320/ST6400 lineages, sharing highly conserved PBP alleles and MDR gene profiles. These findings suggest that the recent resurgence of 19A may be largely associated with the clonal expansion of stable, high-risk MDR lineages. However, given the lack of vaccination history data, we cannot fully exclude the possibility that a proportion of these cases may represent vaccine breakthrough infections.

**Table 3. Virulence genes identified in serotype 19A isolates of *S. pneumoniae*.**

| Genes | | Isolates | | | | | | | | | | |
|---|---|---|---|---|---|---|---|---|---|---|---|---|
| | | 18-85 | 18-185 | 20-1 | 23-4 | 23-71 | 23-113 | 24-43 | 24-175 | 24-189 | 24-243 | 24-250 |
| **Adhesion** | cbpD | + | + | + | + | + | + | + | + | + | + | + |
| | cbpG | + | + | + | + | + | + | + | + | + | + | + |
| | lytA | + | + | + | + | + | + | + | + | + | + | + |
| | lytB | + | + | + | + | + | + | + | + | + | + | + |
| | lytC | + | + | + | + | + | + | + | + | + | + | + |
| | pce/cbpE | + | + | + | + | + | + | + | + | + | + | + |
| | pspA | + | + | + | + | + | + | + | + | + | + | + |
| | pspC/cbpA | – | + | – | + | – | – | + | – | – | + | + |
| | pavA | + | + | + | + | + | + | + | + | + | + | + |
| | lmb | + | + | + | + | + | + | + | + | + | + | + |
| | slrA | + | + | + | + | + | + | + | + | + | + | + |
| | plr/gapA | + | + | + | + | + | + | + | + | + | + | + |
| | rrgA | + | + | + | + | + | + | + | + | + | + | + |
| | rrgB | + | + | + | + | + | + | + | + | + | + | + |
| | rrgC | + | + | + | + | + | + | + | + | + | + | + |
| | srtB | + | + | + | + | + | + | + | + | + | + | + |
| | srtC | + | + | + | + | + | + | + | + | + | + | + |
| | srtD | + | + | + | + | + | + | + | + | + | + | + |
| **Enzyme** | hysA | + | + | + | + | + | + | + | + | + | + | + |
| | nanA | + | + | + | + | + | + | + | + | + | + | + |
| | eno | + | + | + | + | + | + | + | + | + | + | + |
| **Iron uptake** | piaA | + | + | + | + | + | + | + | + | + | + | + |
| | piuA | + | + | + | + | + | + | + | + | + | + | + |
| **Manganese uptake** | psaA | + | + | + | + | + | + | + | + | + | + | + |
| **Protease** | cppA | + | + | + | + | + | + | + | + | + | + | + |
| | iga | + | + | + | + | + | + | + | + | + | + | + |
| | htrA/degP | + | + | + | + | + | + | + | + | + | + | + |
| | tig/ropA | + | + | + | + | + | + | + | + | + | + | + |
| | zmpB | + | + | + | + | + | + | + | + | + | + | + |
| **Toxin** | ply | + | + | + | + | + | + | + | + | + | + | + |

The present table shows the absence (-) or presence (+) of each considered gene from the Virulence Factors database using VFDB.

A particularly notable finding was the shift in β-lactam resistance. While historical ST320 isolates often exhibit high-level cefotaxime resistance [21], the isolates in this study displayed consistent resistance to meropenem with conserved susceptibility to cefotaxime. This resistance phenotype appears to be correlated with a specific combination of pbp alleles pbp1a-13, pbp2b-11, and pbp2x-16, which encode amino acid substitutions in the transpeptidase domain that reduce the binding affinity for carbapenems while maintaining activity against third-generation cephalosporins [15]. Similar PBP profiles have been reported in ST320 isolates in East Asia, supporting the hypothesis that carbapenem resistance may evolve through localized adaptive processes under selective pressure [8]. Moreover, the majority of the isolates harbored multiple resistance determinants, including *erm*(B), *mdf*(A), *msr*(D), and *tet*(M), which is consistent with the fixed MDR genotype within the GPSC1-CC320 lineage. This genetic stability may contribute to the clonal persistence and dissemination of these strains, even in highly vaccinated populations [22].

**Fig 2. Phylogenetic tree of 11 serotype 19A isolates of *S. pneumoniae* clustered with 1300 isolates of GPSC1.**

The re-emergence of ST320 and ST6400 despite the inclusion of serotype 19A in PCV13 cannot be attributed solely to incomplete vaccine coverage. Capsular switching between serotypes 19F and 19A has been well documented both locally and internationally [22] and could explain the continued circulation of vaccine-included clones. Comparative genomic analysis further revealed high similarity between our isolates and reference strains from Japan, Peru, and North America, suggesting a combination of local persistence and cross-border transmission events [20,22]

These findings highlight several implications for pneumococcal disease control. First, the reappearance of high-risk clones within vaccine-covered serotypes supports the consideration of higher-valency conjugate vaccines such as PCV20 in national immunization programs. Second, the emergence of meropenem resistance in pediatric isolates with cefotaxime susceptibility raises important questions regarding empiric therapy and β-lactam stewardship. Third, the observed clonal dynamics underscore the need for genome-based surveillance strategies that go beyond serotype-based classification and allow early detection of clonal shifts.

This study has some limitations. First, pneumococcal serotype analysis was recommended but not mandated under the Infectious Diseases Surveillance Systems, and thus the data may not fully represent the entire population. Consequently, these findings should be interpreted with caution, as this limitation may affect the interpretation of surveillance trends and subsequent analyses.

Second, due to the lack of individual vaccination history data, we were unable to fully evaluate the contribution of vaccine breakthrough infections to the resurgence of serotype 19A. Future studies incorporating detailed vaccination records will be essential to better clarify the relative roles of clonal expansion and vaccine-induced immunity in the re-emergence of serotype 19A.

To our knowledge, this is the first study to characterize meropenem-resistant serotype 19A GPSC1 clones in in children from South Korea at the genomic level. These findings provide further insight into pneumococcal evolution under vaccine

and antibiotic pressure. Future research incorporating clinical metadata, including data on vaccination status, transmission networks, and clinical outcomes, will be essential for comprehensively elucidating the public health impact of these re-emergent MDR clones.

## Author contributions

**Conceptualization:** Junyoung Kim.

**Data curation:** Jin Lee, Jungsun Park, Tae-Min La, Ji-Yeon Hyeon, Dong Hyeok Kim.

**Formal analysis:** Jin Lee, Jungsun Park, Tae-Min La, Ji-Yeon Hyeon.

**Investigation:** Jin Lee, Hwanhee Kim, Seo Yeon Ko.

**Project administration:** Jaeil Yoo, Junyoung Kim.

**Supervision:** Junyoung Kim.

**Visualization:** Jin Lee, Tae-Min La, Ji-Yeon Hyeon.

**Writing – original draft:** Jin Lee, Jungsun Park.

**Writing – review & editing:** Jin Lee, Jungsun Park, Junyoung Kim.

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
