## [Decision Letter · Decision Letter 0]

19 Aug 2025

PONE-D-25-27201Genomic and Epidemiological Dynamics of Meropenem-Resistant GPSC1-CC320 Streptococcus pneumoniae Serotype 19A Isolates from Children Under 5 Years of Age with Invasive Infections, 2018–2024PLOS ONE

Dear Dr. Kim,

Thank you for submitting your manuscript to PLOS ONE. After careful consideration, we feel that it has merit but does not fully meet PLOS ONE’s publication criteria as it currently stands. Therefore, we invite you to submit a revised version of the manuscript that addresses the points raised during the review process. Reviewer 1 has provided an excellent and succinct appraisal, and the identified concerns would require only relatively minor modifications to the manuscript. I fully agree with the reviewer's assessment, and look forward to receiving the revised manuscript from the authors.

We look forward to receiving your revised manuscript.

Kind regards,

Herman Tse

Academic Editor

PLOS ONE

Journal Requirements:

This work was supported by a grant from the Korea Disease Control and Prevention Agency (grant number 6331-301-210)

3. Please ensure that you refer to Figure 1 in your text as, if accepted, production will need this reference to link the reader to the figure.

Reviewers' comments:

Reviewer's Responses to Questions

**Comments to the Author**

1. Is the manuscript technically sound, and do the data support the conclusions?

Reviewer #1: Partly

2. Has the statistical analysis been performed appropriately and rigorously? 

Reviewer #1: No

3. Have the authors made all data underlying the findings in their manuscript fully available?

Reviewer #1: Yes

4. Is the manuscript presented in an intelligible fashion and written in standard English?

Reviewer #1: Yes

5. Review Comments to the Author

Reviewer #1: This study highlights a notable reemergence of serotypes 19A prevalence and disease post-PCV introduction, provides explanation of resistance mechanisms, and potential reasoning behind the increase in 19A prevalence. All of which highlights the importance to monitor this specific serotype in the South Korean children population. Overall, the manuscript reads well and is clearly written. There is originality and significance to the dataset, however, additional issues (major and minor) need to be addressed.

Major issues

1. The study claims “19A is driven by stable, high-risk MDR clones rather than vaccine failure alone.” However, the evidence for this is not very convincing e.g. besides the evidence that these 19A isolates resembles those that are found in other parts of the world – is it difficult to exclude that these 19A disease isolates are not a result of breakthrough infections.

2. Line 150-152: “markedly increased in recent years…rate increasing significantly” - was there stat significant difference in the increase? If so, there’s no mention of what stats were used to test significance. Are these results supported by figure 1, if so, cite figure 1.

3. Line 192: Generally, the discussion is lacking e.g. it is advisable to include a limitations section to the study.

Minor issues

4. Line 54: write out acronym “PSERENADE”

5. Line 63: what are examples of region-specific adaptations?

6. Line 66: why report on uptake rate and not coverage rate?

7. Line 70: sometimes just “Korea” is written. Try to be consistent and use “South Korea” and ” South Korean” etc throughout the whole manuscript.

8. Line 84: of the 104 isolates, what were the distribution of the isolates over the years?

9. Line 89: “species” – clarify what you mean by species e.g. presence of S pneumo

10. Line 91: “after confirmation” – clarify what you mean by this e.g. confirmation of presence of S pneumo

11. Line 107: how many samples were CSF and nonCSF?

12. Line 135: write out acronym for “KDCA” and “IRB”

13. Line 155: “tested” – change to “were tested”

14. Line 156: “All the isolates...” – change to “All 11 isolates...”

15. Line 164-166: are you implying that the presence of that specific PBP type combination explains the intermediate or resistance to those antibiotics? If so, is there evidence/reference that supports this?

16. Line 178: “The GPSC 1 isolates” – clarify that the study isolates here

17. Line 179: “3, 6C, 14, 19A, 19F and 23F serotypes” – change to “serotypes 3, 6C, 14, 19A, 19F and 23F”

18. Line 200: “No. 19A” – what do you mean? The number of 19A isolates?

6. PLOS authors have the option to publish the peer review history of their article (what does this mean? ). If published, this will include your full peer review and any attached files.

**Do you want your identity to be public for this peer review?** For information about this choice, including consent withdrawal, please see our Privacy Policy .

Reviewer #1: No

---

## [Author Response · Author response to Decision Letter 1]

14 Sep 2025

Responses to reviewer’s comments

[Manuscript ID: PONE-D-25-27201]

Genomic and Epidemiological Dynamics of Meropenem-Resistant GPSC1-CC320

Streptococcus pneumoniae Serotype 19A Isolates from Children Under 5 Years of Age

with Invasive Infections, 2018–2024

● Reviewer #1

1. The study claims “19A is driven by stable, high-risk MDR clones rather than vaccine failure alone.” However, the evidence for this is not very convincing e.g. besides the evidence that these 19A isolates resembles those that are found in other parts of the world – is it difficult to exclude that these 19A disease isolates are not a result of breakthrough infections.

- Response: We sincerely appreciate the reviewer’s insightful comment. We agree that, due to the lack of individual vaccination history data in our dataset, it is not possible to exclude the potential contribution of vaccine breakthrough infections to the recent increase in serotype 19A invasive pneumococcal disease (IPD). To address this concern, we have revised the Discussion section to clarify our interpretation and provide a balanced perspective on the role of clonal expansion versus vaccine breakthrough infections:

Revised text (Discussion, Page 11, Lines 218–223):

“Furthermore, our phylogenetic analysis demonstrated that South Korean 19A isolates formed four distinct subclades interspersed within international ST320/ST6400 lineages, sharing highly conserved PBP alleles and MDR gene profiles. These findings suggest that the recent resurgence of 19A may be largely associated with the clonal expansion of stable, high-risk MDR lineages. However, given the lack of vaccination history data, we cannot fully exclude the possibility that a proportion of these cases may represent vaccine breakthrough infections.”

Additionally, we have reinforced this point in the Limitations section (Page 13, Lines 260–263) by noting that the absence of vaccination history data limited our ability to fully evaluate the contribution of vaccine breakthrough infections. We also emphasize that future studies incorporating detailed vaccination records will be essential to clarify the relative contributions of clonal expansion and vaccine-induced immunity in the re-emergence of serotype 19A.

2. Line 150-152: “markedly increased in recent years…rate increasing significantly” - was there stat significant difference in the increase? If so, there’s no mention of what stats were used to test significance. Are these results supported by figure 1, if so, cite figure 1.

- Response: We appreciate the reviewer’s valuable comment. As noted, the original manuscript did not sufficiently specify the statistical methods used to evaluate the observed differences. To address this, we have clarified that the statistical analyses were performed using Fisher’s exact test, provided the corresponding p-values, and cited Figure 1 to indicate that the results are visually supported. We have also updated the Figure 1 caption to include the statistical significance.

Revised text (Results, Page 8, Lines 157–162):

“The proportion of PCV13 serotypes has markedly increased in recent years, increasing from 9.1% (2/22) in 2018 to 27.8% (5/18) in 2023 and reaching 31.6% (6/19) in 2024. Notably, serotype 19A was the most predominant PCV13 serotype, with its detection rate increasing from 9.1% (2/22) in (2018) and 14.3% (1/7) in (2020) to 16.7% (3/18) in (2023) and 26.3% (5/19) in (2024). Furthermore, a comparison between the two periods (2018–2022 vs. 2023–2024) demonstrated significant increases in both the overall proportion of PCV13 serotypes (p = 0.0023) and serotype 19A specifically (p = 0.0154) (Fig 1).”

Revised Figure 1 caption:

“Fig 1. Distribution of PCV13 and non-PCV13 serotypes among invasive S. pneumoniae isolates from Korean children under 5 years of age, 2018–2024.

(A) Annual distribution of serotype 19A among 104 S. pneumoniae isolates from children <5 years of age with IPD in South Korea, 2018–2024.

(B) Comparison of PCV13 versus non-PCV13 isolates (left) and serotype 19A versus non-19A isolates (right) between the periods 2018–2022 and 2023–2024. Statistical significance was determined using Fisher’s exact test (left: p = 0.0023; right: p = 0.0154)”

3. Line 192: Generally, the discussion is lacking e.g. it is advisable to include a limitations section to the study.

- Response: We thank the reviewer for this important suggestion. In response, we have added a dedicated Limitations section to the manuscript to clearly outline the key constraints of our study:

New Limitations Section (Page 12, Lines 256–263):

“This study has several limitations. First, pneumococcal serotype analysis was recommended but not mandated under the Infectious Diseases Surveillance Systems, and thus the data may not fully represent the entire population. Consequently, these findings should be interpreted with caution, as this limitation may affect the interpretation of surveillance trends and subsequent analyses.

Second, due to the lack of individual vaccination history data, we were unable to fully evaluate the contribution of vaccine breakthrough infections to the resurgence of serotype 19A. Future studies incorporating detailed vaccination records will be essential to better clarify the relative roles of clonal expansion and vaccine-induced immunity in the re-emergence of serotype 19A.”

4. Line 54: write out acronym “PSERENADE”

- Response: Thank you for pointing this out. We have written out the full name of PSERENADE upon its first mention in the Introduction (Page 3, Line 54-55).

5. Line 63: what are examples of region-specific adaptations?

- Response: We appreciate the reviewer’s insightful suggestion. We have clarified the text by providing specific examples of region-specific adaptations, including variations in β-lactam resistance-associated pbp alleles and recombination events leading to capsular switching [8,10], and added Reference [10] to support this statement (Page 4, Lines 64–65).

10. Puzia W, Gawor J, Gromadka R, Żuchniewicz K, Wróbel-Pawelczyk I, et al. Highly resistant serotype 19A Streptococcus pneumoniae of the GPSC1/CC320 clone from invasive infections in Poland prior to antipneumococcal vaccination of children. Infect Dis Ther. 2023;12:2017–37. https://doi.org/10.1007/s40121-023-00842-w

6. Line 66: why report on uptake rate and not coverage rate?

- Response: We are grateful for the reviewer’s comment. The value of 88.6% was cited directly from Reference [11], which reports that “88.6% of infants are vaccinated with PCV13.” To maintain consistency and accuracy, we used the term “uptake rate” in the manuscript when referring to this data (Page 4, Lines 68).

7. Line 70: sometimes just “Korea” is written. Try to be consistent and use “South Korea” and ” South Korean” etc throughout the whole manuscript.

- Response: Thank you for the suggestion. We have thoroughly reviewed the manuscript and revised all instances to consistently use “South Korea

8. Line 84: of the 104 isolates, what were the distribution of the isolates over the years?

- Response: We appreciate the reviewer’s helpful suggestion. We have added the annual distribution of the 104 isolates in the Methods section for clarity: 2018 (22), 2019 (10), 2020 (7), 2021 (13), 2022 (15), 2023 (18), and 2024 (19) (Page 5, Lines 87–89).

9. Line 89: “species” – clarify what you mean by species e.g. presence of S pneumo

- Response: We have clarified that “species” refers to the confirmation of the presence of S. pneumoniae (Page 5, Line 92).

10. Line 91: “after confirmation” – clarify what you mean by this e.g. confirmation of presence of S pneumo

- Response: We appreciate the reviewer’s comment. We have revised the text by replacing “After confirmation” with “After confirming the identification of S. pneumoniae” to clarify that the confirmation refers specifically to the identification of S. pneumoniae (Page 5, Line 94).

11. Line 107: how many samples were CSF and nonCSF?

- Response: We thank the reviewer for this helpful comment. We have added the following information to the Methods section for clarification:

“Among the 11 serotype 19A isolates, one isolate was recovered from cerebrospinal fluid (CSF), and the remaining ten were from non-CSF sources.” (Page 6, Lines 111–112).

12. Line 135: write out acronym for “KDCA” and “IRB”

- Response: We thank the reviewer for this helpful comment. The full name of KDCA (Korea Disease Control and Prevention Agency) was already written out at its first mention in the Methods section. For IRB, we have added its full name (Institutional Review Board) at its first occurrence, as suggested. (Page 7, Lines 143)

13. Line 155: “tested” – change to “were tested”

- Response: Corrected as suggested (Page 8, Line 164).

14. Line 156: “All the isolates...” – change to “All 11 isolates...”

- Response: Revised as suggested (Page 8, Line 165).

15. Line 164-166: are you implying that the presence of that specific PBP type combination explains the intermediate or resistance to those antibiotics? If so, is there evidence/reference that supports this?

- Response: We thank the reviewer for this important point. We have clarified that the observed PBP type combinations are associated with intermediate resistance and non-susceptibility, and added Reference [15] to support this statement (Page 9, Line 175).

15. Varghese R, Basu S, Neeravi A, et al. Emergence of meropenem resistance among cefotaxime non-susceptible Streptococcus pneumoniae: evidence and challenges. Front Microbiol. 2022;12:810414. https://doi.org/10.3389/fmicb.2021.810414

16. Line 178: “The GPSC 1 isolates” – clarify that the study isolates here

- Response: We thank the reviewer for the helpful suggestion. To clarify that the isolates refer specifically to those identified in this study, we have revised the text to:

“The GPSC1 isolates identified in this study” (Page 9, Line 187).

17. Line 179: “3, 6C, 14, 19A, 19F and 23F serotypes” – change to “serotypes 3, 6C, 14, 19A, 19F and 23F”

- Response: Revised as suggested (Page 10, Line 188).

18. Line 200: “No. 19A” – what do you mean? The number of 19A isolates?

- Response: We thank the reviewer for this helpful comment. To improve clarity, we have revised the sentence to:

“Notably, serotype 19A was not detected.” (Page 10, Line 209).

This concludes our response to the reviewer’s comments. We believe they have helped us improve our work considerably. We hope that they now find our manuscript acceptable for publication.

---

## [Decision Letter · Decision Letter 1]

17 Nov 2025

PONE-D-25-27201R1Genomic and Epidemiological Dynamics of Meropenem-Resistant GPSC1-CC320 Streptococcus pneumoniae Serotype 19A Isolates from Children Under 5 Years of Age with Invasive Infections, 2018–2024PLOS ONE

Dear Dr. Kim,

Thank you for submitting your manuscript to PLOS ONE. After careful consideration, we feel that it has merit but does not fully meet PLOS ONE’s publication criteria as it currently stands. Therefore, we invite you to submit a revised version of the manuscript that addresses the points raised during the review process. One of the reviewers still have issues with the manuscript that needs to be addressed carefully.  Please see reviewer's insightful comments below.

We look forward to receiving your revised manuscript.

Kind regards,

Baochuan Lin, Ph.D.

Academic Editor

PLOS ONE

Journal Requirements:

Reviewers' comments:

Reviewer's Responses to Questions

**Comments to the Author**

1. If the authors have adequately addressed your comments raised in a previous round of review and you feel that this manuscript is now acceptable for publication, you may indicate that here to bypass the “Comments to the Author” section, enter your conflict of interest statement in the “Confidential to Editor” section, and submit your "Accept" recommendation.

Reviewer #1: All comments have been addressed

Reviewer #2: All comments have been addressed

2. Is the manuscript technically sound, and do the data support the conclusions?

Reviewer #1: Yes

Reviewer #2: Partly

3. Has the statistical analysis been performed appropriately and rigorously? 

Reviewer #1: Yes

Reviewer #2: N/A

4. Have the authors made all data underlying the findings in their manuscript fully available?

Reviewer #1: Yes

Reviewer #2: Yes

5. Is the manuscript presented in an intelligible fashion and written in standard English?

Reviewer #1: Yes

Reviewer #2: Yes

6. Review Comments to the Author

Reviewer #1: All previous commends and suggestions have been addressed sufficiently by the authors including the addition of the study's limitations and statistical analysis.

Reviewer #2: The authors characterized a set of isolates of S. pneumoniae belonging to serotype 19A recovered from children under aged less than 5 years in South Korea between 2018 and 2024. They used whole-genome sequencing and analyzed clonal composition, resistance profiles, and phylogenetic relationships of the isolates to assess the genomic and epidemiologic factors contributing to their re-emergence. This work is of great importance, including in global level, because serotype 19A is prevalent in multiple regions worldwide and is frequently associated with antimicrobial resistance. Genomic surveillance of these isolates may provide critical insights into clonal dissemination, vaccine impact, and resistance dynamics, thereby informing international public health strategies. However, a few points need to be clarified.

In the abstract, the authors state the “Our findings suggest that the resurgence of 19A is driven by stable, high-risk MDR clones rather than vaccine failure alone”. The data provided by the authors did not allow them to suggest that vaccine failure happened or not. So, please remove “rather than vaccine failure alone”.

Vaccine uptake and vaccine coverage may not have the same meaning. In the Introduction (line 68), the statement “… with PCV13 achieving an uptake rate of 88.6%” is not very clear. Is it the proportion of vaccinated children who took PCV13? If so, it does not reflect vaccine coverage. The authors need to show how the vaccine coverage was overtime, since the data they referenced are from 10 years ago. If no newer information about vaccine coverage, especially with PCV13, is available, the authors need to make it clear.

In the objective, the authors state that they will assess genomic and epidemiologic factors contributing to the re-emergence of serotype 19A. However, I did not find any epidemiological aspects discussed in this manuscript. Nothing related to the population was compared. I recommend removing the term “epidemiologic” from here and the abstract.

Methods (lines112-113): “For isolates obtained from cerebrospinal fluid (CSF)…” Only one isolate was recovered from CSF. So, rephrase it. “For the single isolate recovered…”

Discussion (first paragraph): “These isolates, belonging to the GPSC1‒CC320 clonal complex, were analyzed to elucidate their resistance mechanisms and the epidemiological context of their re-emergence in a population with high PCV13 coverage”. Again, no classical epidemiological data were available in the study. Epidemiological data refer to the characteristics of the population from which the isolates were obtained. Vaccination history of the children was not available, for instance. So, this term should be removed.

Also, I did not see any updated information on PCV13 coverage in the study. To keep this statement at the end of the first paragraph of the discussion, the authors need to include in the introduction session the vaccine coverage over the years of study or at least the most recent data available. At least, the PCV13 coverage during the study period needs to be stated in the introduction.

Limitations: change “several” to “some”. Also, in general, a paragraph with limitations comes before the final paragraph (conclusions). Check that.

7. PLOS authors have the option to publish the peer review history of their article (what does this mean? ). If published, this will include your full peer review and any attached files.

**Do you want your identity to be public for this peer review?** For information about this choice, including consent withdrawal, please see our Privacy Policy .

Reviewer #1: No

Reviewer #2: **Yes: ** FELIPE PIEDADE GONÇALVES NEVES

---

## [Author Response · Author response to Decision Letter 2]

24 Nov 2025

Responses to reviewer’s comments

[Manuscript ID: PONE-D-25-27201]

Genomic Insights into the Expansion of Meropenem-Resistant GPSC1-CC320 Streptococcus pneumoniae Serotype 19A Isolates from Children Under 5 Years of Age with Invasive Infections, 2018–2024

● Reviewer #2

1. In the abstract, the authors state the “Our findings suggest that the resurgence of 19A is driven by stable, high-risk MDR clones rather than vaccine failure alone”. The data provided by the authors did not allow them to suggest that vaccine failure happened or not. So, please remove “rather than vaccine failure alone”.

- Response: Thank you for this helpful comment. We agree that our data do not support any conclusion regarding vaccine failure. To avoid suggesting an interpretation that cannot be substantiated by our dataset, we removed the phrase “rather than vaccine failure alone” from the abstract. (Abstract, Page 2, Line 36)

2. Vaccine uptake and vaccine coverage may not have the same meaning. In the Introduction (line 68), the statement “… with PCV13 achieving an uptake rate of 88.6%” is not very clear. Is it the proportion of vaccinated children who took PCV13? If so, it does not reflect vaccine coverage. The authors need to show how the vaccine coverage was overtime, since the data they referenced are from 10 years ago. If no newer information about vaccine coverage, especially with PCV13, is available, the authors need to make it clear.

- Response: Thank you for raising this important point. We agree that the previously cited PCV13 uptake value was outdated and that the terminology used lacked clarity. To address this, we removed the outdated uptake figure and revised the Introduction to incorporate the most recent national immunization coverage data. “According to the latest national report, PCV coverage among children in South Korea reached 97% in 2023” and the corresponding reference has been updated accordingly. (Introduction, Page 4, Lines 68–69)

3. In the objective, the authors state that they will assess genomic and epidemiologic factors contributing to the re-emergence of serotype 19A. However, I did not find any epidemiological aspects discussed in this manuscript. Nothing related to the population was compared. I recommend removing the term “epidemiologic” from here and the abstract.

- Response: Thank you for this helpful observation. Our study does not include population-level epidemiological comparisons, and the use of the term “epidemiologic” could therefore be misleading. We carefully reviewed the manuscript and removed the term from all relevant sections, including the Title, Introduction, Method, and Discussion (Title; Introduction, Page 4, Line 77 and 81; Methods, Page 7, Line 140; Discussion, Page 10, Line 207)

4. Methods (lines 112-113): “For isolates obtained from cerebrospinal fluid (CSF)…” Only one isolate was recovered from CSF. So, rephrase it. “For the single isolate recovered…”

- Response: Thank you for pointing this out. We revised the sentence to accurately indicate that only one isolate was obtained from CSF. (Methods, Page 6, Line 114)

5. Discussion (first paragraph): “These isolates, belonging to the GPSC1‒CC320 clonal complex, were analyzed to elucidate their resistance mechanisms and the epidemiological context of their re-emergence in a population with high PCV13 coverage”. Again, no classical epidemiological data were available in the study. Epidemiological data refer to the characteristics of the population from which the isolates were obtained. Vaccination history of the children was not available, for instance. So, this term should be removed.

- Response: We appreciate your clarification on this point. As our study does not include population-level epidemiological data, we removed the term “epidemiological context” from the Discussion and revised the sentence to focus solely on genomic and resistance-related findings. (Discussion, Page 10, Lines 207)

6. Also, I did not see any updated information on PCV13 coverage in the study. To keep this statement at the end of the first paragraph of the discussion, the authors need to include in the introduction session the vaccine coverage over the years of study or at least the most recent data available. At least, the PCV13 coverage during the study period needs to be stated in the introduction.

- Response: Your comment appropriately underscores the need updated vaccine coverage information. In response, we revised the Introduction to include the most recent national data indicating that PCV coverage among children in South Korea reached 97% in 2023, and updated the corresponding reference (Introduction, Page 4, Lines 68–69).

7. Limitations: change “several” to “some”. Also, in general, a paragraph with limitations comes before the final paragraph (conclusions).

- Response: We acknowledge your guidance regarding the structure of the Discussion section. Following your recommendation, we replaced “several” with “some” and repositioned the Limitations paragraph to appear immediately before the final summary paragraph (Discussion, Page 12, Lines 251–258).

This concludes our response to the reviewer’s comments. We believe that the revisions have substantially strengthened the clarity and scientific rigor of the manuscript. We appreciate the reviewer’s thoughtful assessment and hope that the revised version will be found suitable for publication.

---

## [Decision Letter · Decision Letter 2]

1 Dec 2025

Genomic Insights into the Expansion of Meropenem-Resistant GPSC1-CC320 Streptococcus pneumoniae Serotype 19A Isolates from Children Under 5 Years of Age with Invasive Infections, 2018–2024

PONE-D-25-27201R2

Dear Dr. Kim,

We’re pleased to inform you that your manuscript has been judged scientifically suitable for publication and will be formally accepted for publication once it meets all outstanding technical requirements.

Kind regards,

Baochuan Lin, Ph.D.

Academic Editor

PLOS ONE

Additional Editor Comments (optional):

Reviewers' comments:

Reviewer's Responses to Questions

**Comments to the Author**

1. If the authors have adequately addressed your comments raised in a previous round of review and you feel that this manuscript is now acceptable for publication, you may indicate that here to bypass the “Comments to the Author” section, enter your conflict of interest statement in the “Confidential to Editor” section, and submit your "Accept" recommendation.

Reviewer #2: All comments have been addressed

2. Is the manuscript technically sound, and do the data support the conclusions?

Reviewer #2: Yes

3. Has the statistical analysis been performed appropriately and rigorously? 

Reviewer #2: N/A

4. Have the authors made all data underlying the findings in their manuscript fully available?

Reviewer #2: Yes

5. Is the manuscript presented in an intelligible fashion and written in standard English?

Reviewer #2: Yes

6. Review Comments to the Author

Reviewer #2: The authors have satisfactorily addressed all of my comments. Therefore, I recommend that the manuscript be accepted.

7. PLOS authors have the option to publish the peer review history of their article (what does this mean? ). If published, this will include your full peer review and any attached files.

**Do you want your identity to be public for this peer review?** For information about this choice, including consent withdrawal, please see our Privacy Policy .

Reviewer #2: **Yes: ** FELIPE PIEDADE GONÇALVES NEVES

---

## [Editor Report · Acceptance letter]

PONE-D-25-27201R2

PLOS One

Dear Dr. Kim,

I'm pleased to inform you that your manuscript has been deemed suitable for publication in PLOS One. Congratulations! Your manuscript is now being handed over to our production team.

Kind regards,

on behalf of

Dr. Baochuan Lin

Academic Editor

PLOS One